

# Blood lactate dynamics in awake and anaesthetized mice after intraperitoneal and subcutaneous injections of lactate—sex matters

Øyvind P. Haugen[1], Evan M. Vallenari[1], Imen Belhaj[1,2], Milada Cvancarova Småstuen[3], Jon Storm-Mathisen[2], Linda H. Bergersen[1] and Ingrid Åmellem[1]

[1] The Brain and Muscle Energy Group, Electron Microscopy Laboratory, Institute of Oral Biology, University of Oslo, Oslo, Norway
[2] Amino Acid Transporter Laboratory, Division of Anatomy, Department of Molecular Medicine, Institute of Basic Medical Sciences, SERTA: Healthy Brain Ageing Centre, University of Oslo, Oslo, Norway
[3] Department of Nursing and Health Promotion, Faculty of Health Science, Oslo Metropolitan University, Oslo, Norway

## ABSTRACT

Lactate treatment has shown a therapeutic potential for several neurological diseases, including Alzheimer's disease. In order to optimize the administration of lactate for studies in mouse models, we compared blood lactate dynamics after intraperitoneal (IP) and subcutaneous (SC) injections. We used the 5xFAD mouse model for familial Alzheimer's disease and performed the experiments in both awake and anaesthetized mice. Blood glucose was used as an indication of the hepatic conversion of lactate. In awake mice, both injection routes resulted in high blood lactate levels, mimicking levels reached during high-intensity training. In anaesthetized mice, SC injections resulted in significantly lower lactate levels compared to IP injections. Interestingly, we observed that awake males had significantly higher lactate levels than awake females, while the opposite sex difference was observed during anaesthesia. We did not find any significant difference between transgenic and wild-type mice and therefore believe that our results can be generalized to other mouse models. These results should be considered when planning experiments using lactate treatment in mice.

## INTRODUCTION

Physical exercise has beneficial effects on health, both in preventing and treating several diseases. Among these are neurological and psychiatric diseases, and physical exercise may delay the progression of Alzheimer's disease (*Tari et al., 2019*). Blood lactate levels can increase several-fold during high-intensity training (*Goodwin et al., 2007*). Some of the lactate produced during exercise crosses the blood–brain barrier and enters the brain, where it can be metabolized (*Pierre & Pellerin, 2005*; *Van Hall et al., 2009*). In men performing high-intensity interval training, improvement in executive functions correlated with blood lactate levels and cerebral lactate uptake (*Hashimoto et al., 2018*). Systemic lactate

Corresponding authors
Linda H. Bergersen,
l.h.bergersen@odont.uio.no
Ingrid Åmellem,
ingrid.amellem@odont.uio.no

administration has been shown to increase levels of brain-derived neurotrophic factor (BDNF) in both humans and mice at rest and improved learning and memory in mice (*El Hayek et al., 2019*; *Schiffer et al., 2011*). BDNF facilitates neuronal growth, differentiation, survival, and synaptic plasticity, and plays a vital role in the pathophysiology of several neurological disorders (*Mitre, Mariga & Chao, 2017*).

We have previously identified the $G_i$-coupled lactate receptor HCAR1 in the brain. Notably, lactate injections mimicked the effect of high-intensity interval training by increasing the combined neuro-angiotrophic factor VEGF and angiogenesis in the hippocampus of wild-type mice, but not in HCAR1 knockout mice (*Morland et al., 2017*). This effect of HCAR1 activation can potentially delay the progression of vascular dementias and Alzheimer's disease. In addition, lactate has been shown to protect against inflammation through an HCAR1-mediated mechanism (*Hoque et al., 2014*), which may be an important component of the pathology of Alzheimer's disease (*Kinney et al., 2018*). Thus, based on the present knowledge, lactate administration has therapeutic potential for several neurological diseases.

Intraperitoneal (IP) and subcutaneous (SC) injections are common routes for drug delivery in rodents. Even though both injection types are considered parenteral, differences in absorption rate and bioavailability can be expected. Drugs administered through SC injection are absorbed through capillaries in the subcutis and often slower than other parenteral administration routes, which in turn can result in a more sustained effect (*Turner et al., 2011*). IP injection resembles to some extent an oral administration route, where drugs are absorbed from the gastrointestinal tract into the portal vein for delivery to the liver. Drugs injected into the peritoneal cavity are absorbed mainly through the mesenteric vessels that drain into the portal vein. Through monocarboxylate transporters, lactate enters hepatic cells, where it can be metabolized. The lactate can be converted to pyruvate in the liver and used as substrate for the synthesis of glucose through gluconeogenesis (*Brooks, 1986*). Some of the lactate injected IP will therefore undergo hepatic metabolism before reaching the systemic blood flow. Similarly, some of the lactate injected SC or produced and released by skeletal muscles during anaerobic exercise, may eventually be metabolized in the liver.

In order to optimize the administration of lactate for studies of the mouse brain, we compared blood lactate dynamics after IP and SC injections in a mouse model (5xFAD) of Alzheimer's disease and in corresponding wild-type control mice (*Oakley et al., 2006*). Blood glucose was measured to monitor the conversion of lactate to glucose by the liver. Comparison of these injection methods for lactate has to our knowledge not been performed in any mouse model. Injections were performed both on awake and anaesthetized mice, induced by isoflurane, which is the most common volatile anaesthetic agent used in laboratory mice (*Gargiulo et al., 2012*). These two conditions might result in different response curves.

## MATERIALS AND METHODS

### Animals

The mice used in this study (5xFAD) (*Oakley et al., 2006*) were obtained from the Mutant Mouse Resource & Research Centers (MMRRC strain name B6SJL-Tg(APPSwFlLon, PSEN1*M146L*L286V)6799Vas/Mmjax; stock number 034840-JAX). They were bred and treated in accordance with the national and regional ethical guidelines. All experiments were performed by certified personnel and approved by the Animal Use and Care Committee of the Institute of Basic Medical Sciences, The Faculty of Medicine, University of Oslo, and by the Norwegian Animal Research Authority (FOTS 17448, 17551, 15525). All the mice were housed under a 12:12 h light/dark cycle and had free access to water and chow.

### Group distributions

Genotypes (wild-type and transgenic) were verified by conventional PCR and gel electrophoresis. The mice were randomly divided into different groups based on treatment, with 5–6 mice in each group. Sex and genotype distributions in the different groups did not exceed 2/3 in favour of one or another, and the average age was between 12 and 17 weeks. See Table S1 for a complete overview of the different groups.

### Lactate injections

Mice subjected to lactate injections received sodium L-lactate (71718, Sigma-Aldrich, MO, USA); 2 g/kg body weight; 200 mg/ml dissolved in PBS, pH 7.4; corresponding to 18 mmol/kg, by IP injection or SC injection on the neck/back (Figs. 1A and 1B). Control mice got the same volume of PBS injected (per kg bodyweight). The concentration of sodium L-lactate mimics lactate levels reached after high-intensity training (*Morland et al., 2017*).

### Anaesthesia

Mice subjected to anaesthesia were exposed to isoflurane in an airflow chamber with 400 ml/min airflow with 4% isoflurane prior to blood samplings and injections (Baxter, San Juan, Puerto Rico). Anaesthesia was maintained with 200 ml/min airflow with 1.7–2.5% isoflurane. A heating block was placed beneath the airflow chamber to maintain a stable body temperature.

### Blood samplings and measurements

Blood samples were taken immediately before (baseline) and at the following time points after injection: 5, 13, 37 min for the awake mice. The time course was extended to include 60 min for mice under anaesthesia to account for potentially delayed absorption and clearance rates. By puncturing the submandibular vein with a five mm long Goldenrod animal lancet (Braintree Scientific, Inc, MA, USA), 20 µl of blood was collected with a capillary tube (Fig. 1C) and immediately emptied into a pre-filled Eppendorf tube with haemolyzing solution (EKF Diagnostics, Cardiff, UK) with subsequent shaking. The samples were analyzed for lactate and glucose levels on a Biosen C-Line GP+ system (EKF Diagnostics, Cardiff, UK). Since it was challenging to collect blood samples at the exact same time points for all animals, we calculated the average time points and used linear

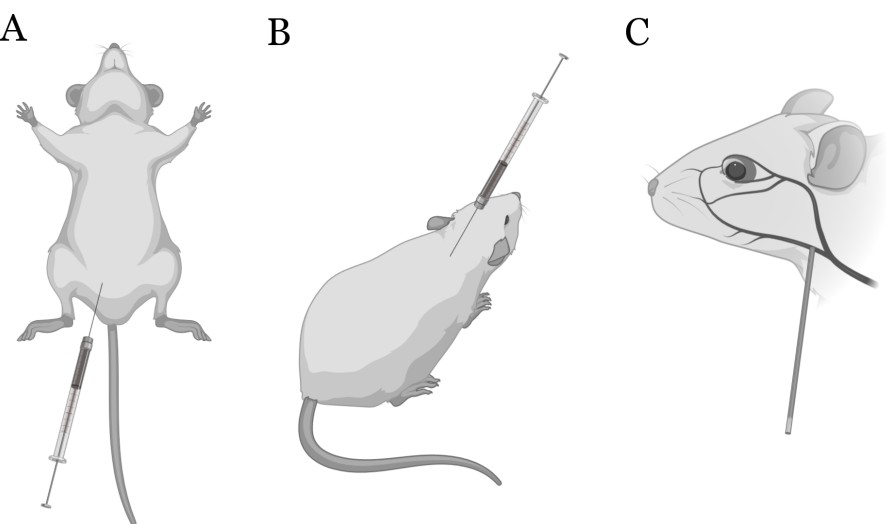

**Figure 1** **Sites of injection and blood sampling.** Intraperitoneal (IP) (A) and subcutaneous (SC) (B) injections in mice. Blood samples were collected from the submandibular vein (C). Illustrated with BioRender.

interpolation to obtain lactate and glucose levels (Fig. 2). The range and average of the actual times are shown in Table S2.

## Statistical analysis

Possible differences between groups were analyzed using a linear mixed effect regression model for repeated measurements. The analyses were performed separately for awake and anaesthetized mice. The model was fitted separately for the two outcomes, lactate and glucose, and the covariates age, sex and genotype were included in the model. In addition, an interaction term "time*group" was fitted to estimate possible differences in time trajectories between the groups. The results are presented as the estimated means with standard error of the regression (S.E.R.). To correct for multiple testing, significance level was set to 0.01. All analyses were performed using Stata ver. 14.2 (StataCorp LLC, TX, USA).

## RESULTS AND DISCUSSION

The baseline levels of blood lactate (group averages ranging from about 6.5 to 9.2 mM in awake mice, Fig. 2A) were higher than expected. Normal physiological blood lactate in mice has previously been reported to be in the range of 2.5 to 4.6 mM (*Iversen et al., 2012*; *Schwarzkopf et al., 2013*). A possible explanation to the high baseline is that the injections, and especially the blood sampling, could be experienced as stressful, causing the lactate levels to rise. This interpretation is supported by the fact that the blood glucose levels continued to rise during the experimental period in the awake mice (Fig. 2C). Both hyperlactataemia and hyperglycaemia are well-known adrenergic responses (*Gjedsted et al., 2011*; *Levy et al., 2003*). In fact, repeated blood sampling with two minute intervals has been

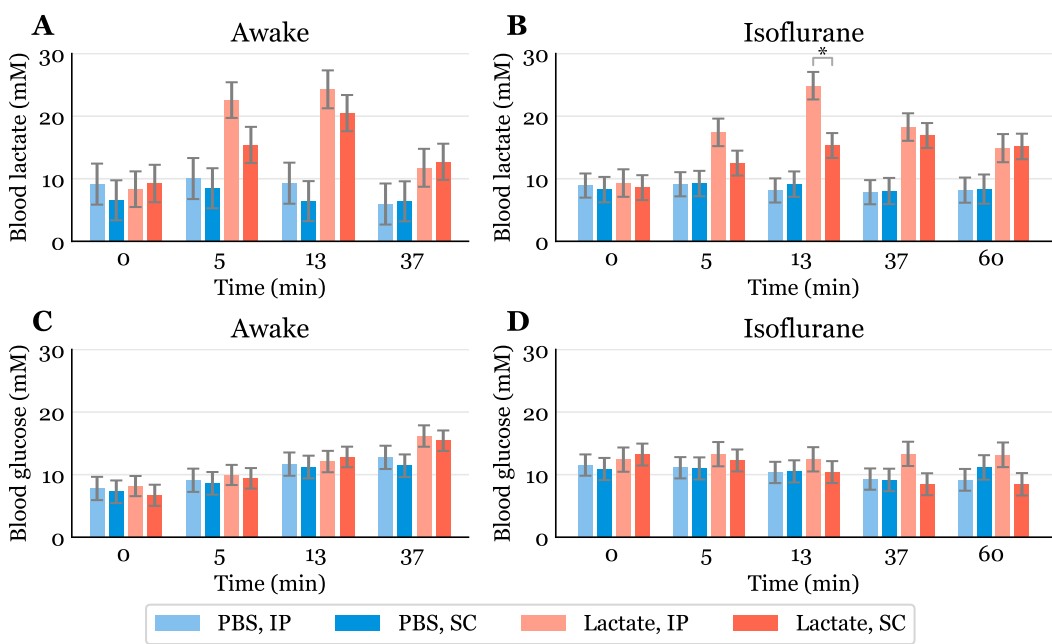

**Figure 2  Time-response for blood lactate and glucose.** Lactate (A, B) and glucose (C, D) measured in blood samples taken from the submandibular vein in awake (A, C) or anaesthetized mice (B, D). Lactate or PBS were injected intraperitoneally (IP) or subcutaneously (SC). Blood samples were taken immediately before (0) and at 5, 13 and 37 min after injection for the awake mice and extended to 60 min for the anaesthetized mice. The data are presented as the estimated mean ± S.E.R., * equals $p < 0.01$. Statistical analyses were done using a linear mixed effect regression model for repeated measurements.

shown to increase plasma glucose in a time-dependent manner in mice (*Tabata, Kitamura & Nagamatsu, 1998*). A stress-induced adrenergic response is expected to be diminished by anaesthesia, but we did not see a change in the baseline lactate levels after isoflurane exposure (Fig. 2B). It has been proposed that isoflurane and other volatile anaesthetics can alter mitochondrial function and thereby affect the synthesis and/or degradation of lactate (*Horn & Klein, 2010*).

In anaesthetized mice, lactate injected IP gave significantly higher lactate levels (24.9 mM, 95% CI [20.6–29.2]) than SC injected (15.3 mM, 95% CI [11.4–19.2]) after 13 min (Fig. 2B). Although we did not find statistically significant differences at other time points in the anaesthetized mice nor in the awake mice at any time point, there is a trend showing higher lactate levels with IP injection than SC injection. This is consistent with the observation that SC injections have the slowest absorption rate of parenteral administration routes (*Turner et al., 2011*).

Since lactate injected IP enters the liver through the portal vein, it was of interest to examine whether lactate was lost by hepatic conversion to glucose. Administration of lactate in rats has been shown to induce hypophagia when injected IP, while no effect on food intake was found in rats receiving SC injections (*Racotta & Russek, 1977*). The authors speculated that this is due to a hepatic action, which probably is mediated by hepatic glucoreceptors. However, we did not find any statistically significant changes in
blood glucose between the groups, although the glucose values were somewhat higher after IP than after SC injections at later time points during anaesthesia (Fig. 2D). We cannot exclude the possibility that some of the lactate was used as a gluconeogenic precursor without affecting blood glucose, which has been shown with $^{13}$C labelled lactate infusions in humans (Miller et al., 2002). Despite this, the high blood lactate levels observed after injections suggest that most of the IP injected lactate reached the systemic blood.

Previous studies have shown that isoflurane induces hyperglycaemia by inhibiting insulin secretion (Schwarzkopf et al., 2013; Zuurbier et al., 2014; Windeløv, Pedersen & Holst, 2016). Our data are consistent with the values reported by Schwarzkopf et al. (2013). In general, different anaesthetics are known to affect metabolism and various physiological characteristics, and it is important to be aware of such confounding effects. In studies that aim to look at the effects of a specific metabolite, the use of anaesthesia should be carefully considered. We used isoflurane in this study, as it is the most common anaesthetic used in mice. Compared to the results obtained from awake mice (Fig. 2A), blood lactate levels after injections during anaesthesia changed at a slower rate, with the highest levels being lower and observed at later time points (Fig. 2C). This agrees with observations of protracted tissue distribution time courses of various markers in isoflurane anaesthesia compared to the awake state (Avram et al., 2000). As the lactate levels seemed to be sustained for a longer time during anaesthesia, we included measurements at 60 min. The levels were approximately twice as high as baseline after 60 min during anaesthesia, while they were closer to baseline after 37 min in the awake mice. The protracted time course of blood lactate after SC compared to IP injection in anaesthetized mice agrees with the observation that blood flow in skin and fat is about one seventh of that in the small intestine of mice anaesthetized with isoflurane (Boswell et al., 2014).

While neither age nor genotype showed statistically significant effects on blood glucose or lactate levels, sex had a significant effect on the lactate levels both during awake and anaesthetized conditions. When awake, male mice had higher lactate levels than female mice by an overall mean difference of 3.3 mM (95% CI [1.1–5.46], $p < 0.01$). During isoflurane-induced anaesthesia, lactate levels in male mice were more than halved, dropping to a lower level than in female mice by an overall mean difference of 3.0 mM (95% CI [1.87–4.1], $p < 0.001$). Lactate levels in female mice did not seem to be notably affected by the isoflurane. The sex differences estimated by the statistical analysis can clearly be seen when presenting the time course for each sex separately, regardless of injection method (Fig. 3).

Sex differences in blood lactate levels have, as far as we know, not been reported in mice, and we can only hypothesize on these findings. A possibility is that, when awake, male mice respond differently to handling than females due to a more aggressive nature. However, in a study by Hurst and West, differences in stress and anxiety responses after routine laboratory handling showed only minor sex differences (Hurst & West, 2010). Differences in muscle physiology could also influence lactate levels, as males generally have more glycolytic muscle fibres than females (Haizlip, Harrison & Leinwand, 2015). Since adrenaline can induce hyperlactataemia (Gjedsted et al., 2011; Levy et al., 2003), it is possible that different muscle fibres could produce different amounts of lactate in response to the

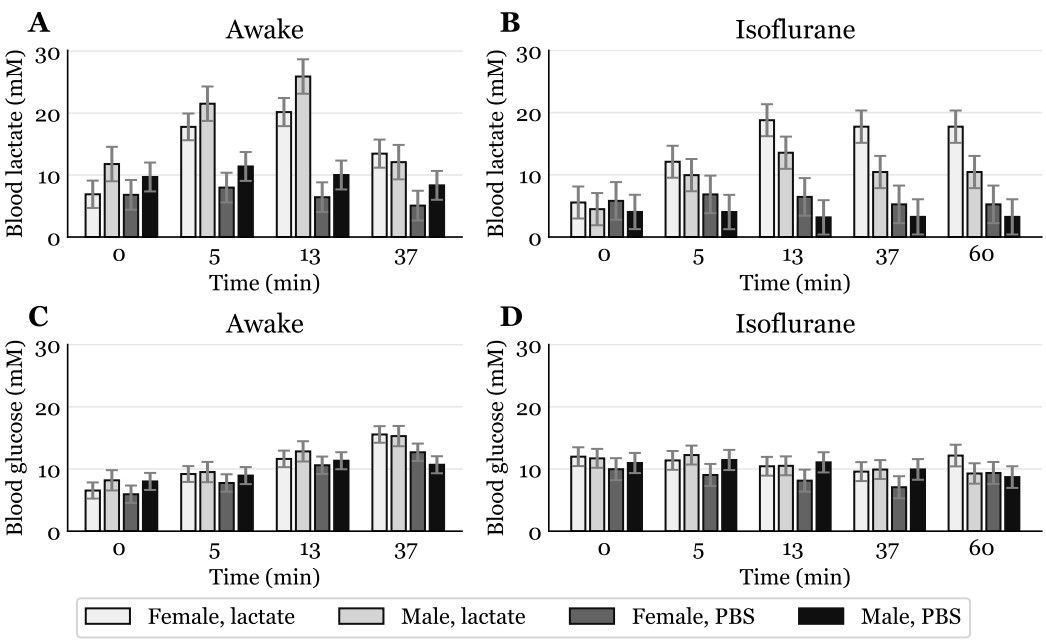

**Figure 3 Time-response for blood lactate and glucose by sex.** Mice that received intraperitoneal (IP) or subcutaneous (SC) injection were pooled, giving one lactate injected group and one PBS injected group for each sex (awake lactate groups: females $n = 7$, males $n = 5$; isoflurane lactate groups: females $n = 6$, males $n = 6$; awake PBS groups: females $n = 5$, males $n = 6$; isoflurane PBS groups: females $n = 5$, males $n = 6$). Lactate (A, B) and glucose (C, D) measured in blood samples taken from the submandibular vein in awake (A, C) or anaesthetized mice (B, D). Blood samples were taken immediately before (0) and at 5, 13 and 37 min after injection for the awake mice and extended to 60 min for the anaesthetized mice. The data are presented as the estimated mean ± S.E.R.

stress of handling and blood sampling. Whatever the reason, the male-to-female difference in mM lactate was present before lactate injections and also occurred after PBS injections in awake mice (Fig. 3A). Different lactate levels in males and females were also observed during anaesthesia, which indicates a sex dependent effect of the anaesthetic. A study performed on rats showed that brief exposure to isoflurane elevates plasma corticosteroid levels in females, but not in males (*Bekhbat et al., 2016*). Such an effect might explain why females have higher lactate levels than males after exposure to isoflurane (Fig. 3B), opposite to what is observed in awake mice. It is possible that elevated stress hormones after exposure to isoflurane prevent lactate levels in females from declining. Future studies including measurements of stress markers are needed to elucidate these sex differences.

## CONCLUSION

Based on our findings, both IP and SC injections are suitable options for administration of lactate in mice. In spite of some differences in the dynamics, both injection routes yielded high blood lactate levels, mimicking levels reached during high-intensity training. When performing lactate injections under isoflurane-induced anaesthesia, one should be aware that SC injections could result in significantly lower lactate levels than IP injections.

There were sex-dependent differences in lactate levels in both awake and anaesthetized conditions. The dynamics, as well as the possible sex-differences, should be taken into account when using mouse models for studies involving lactate treatments.

### Funding

Funding contribution was obtained from the Norwegian Health Association, project ID 14848, project leader Linda H. Bergersen. The funders had no role in study design, data collection and analysis, decision to publish, or preparation of the manuscript.

### Grant Disclosures

The following grant information was disclosed by the authors:
Norwegian Health Association: 14848.

### Competing Interests

The authors declare there are no competing interests.

### Author Contributions

- Øyvind P. Haugen analyzed the data, conceived and designed the experiments, performed the experiments, prepared figures and/or tables, authored or reviewed drafts of the paper, and approved the final draft.
- Evan M. Vallenari conceived and designed the experiments, performed the experiments, authored or reviewed drafts of the paper, and approved the final draft.
- Imen Belhaj conceived and designed the experiments, performed the experiments, authored or reviewed drafts of the paper, and approved the final draft.
- Milada Cvancarova Småstuen analyzed the data, authored or reviewed drafts of the paper, and approved the final draft.
- Jon Storm-Mathisen and Linda H. Bergersen conceived and designed the experiments, authored or reviewed drafts of the paper, and approved the final draft.
- Ingrid Åmellem analyzed the data, conceived and designed the experiments, performed the experiments, prepared figures and/or tables, authored or reviewed drafts of the paper, and approved the final draft.

### Animal Ethics

The following information was supplied relating to ethical approvals (i.e., approving body and any reference numbers):

The Animal Use and Care Committee of the Institute of Basic Medical Sciences, The Faculty of Medicine, University of Oslo, and the Norwegian Animal Research Authority approved this research (FOTS 17448, 17551, 15525).

### Data Availability

The raw data is available in the Supplementary Files.

## Supplemental Information

Supplemental information for this article can be found online at http://dx.doi.org/10.7717/peerj.8328#supplemental-information.

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
