# Peer review of "Blood lactate dynamics in awake and anaesthetized mice after intraperitoneal and subcutaneous injections of lactate—sex matters"

_PeerJ, doi:10.7717/peerj.8328_

## Round 0.1 · original submission · Minor Revisions

While the reviewers were largely positive, a few questions/issues were brought up that should be addressed. In particular, do consider the recommendations of reviewer #2 with regard to the presentation of data in figures 2 and 3.

·

Basic reporting

This is an extremely important finding, especially for rodent studies intended for mimicking humans. The effect of animal handling and the adrenergic response in rodent was very well documented in the 1970 through 1990 literature. This was time when IUCAC would permit decapitation without anesthesia so a true comparison could be determined. I think a more thorough review of the literature would permit the authors to add in the Abstract that "the difference in awake and anesthetized mice is due to the adrenergic response caused by mouse handling. A fairly thorough review of this early literature was performed in a section from a recent review (Tikunov et al, Current Metabolomics, 2013). They should know that most of the hepatic glucose is derived from glycogen in those initial minutes during handling. Without 13C-labeled lactate given SC or IP, it is not possible distinguish glucose from glycogenolysis or gluconeogenesis. In fact, early In vivo 13C MRS glycogen studies infused beta-blockers to lab animals to inhibit glycogenolysis (in Tikunov et al., 2013). The finding that male mice have a significantly elevated response to handling and this could be due to their inherent aggressiveness could be novel but a more thorough literature review is required. They should include some of these earlier references to help substantiate their results, which are taken from the Tikunov et al, 2013 review. Especially important for thier study are the first 4 references and the last 4 books in this list:

Gariépy, J. L.; Rodriguiz, R. M.; Jones, B. C. Handling, genetic and housing effects on the mouse stress system, dopamine function, and behavior. Pharmacol. Biochem. Behav., 2002, 73, 7-17.

Tabata, H.; Kitamura, T.; Nagamatsu, N. Comparison of effects of restraint, cage transportation, anaesthesia and repeated bleeding on plasma glucose levels between mice and rats. Lab. Anim., 1998, 32, 143-148.

Zethof, T. J.; Van der Heyden, J. A.; Tolboom, J. T.; Olivier, B. Stress-induced hyperthermia in mice: A methodological study. Physiol. Behav., 1994, 55, 109-115.

Annamunthodo, H.; Keating, V. J.; Patrick, S. J. Liver glycogen alterations in anaesthesia and surgery. Anaesthesia, 1958, 13, 429- 433.

Besch, E. L.; Chou, B. J. Physiological responses to blood collection methods in rats. PSERM., 1971, 138, 1019-1022.

Arola, L. L.; Palou, A.; Remesar, X.; Herrara, E.; Alemany, L. Effect of stress and sampling site on metabolite concentration in rat plasma. Arch. Internat. Physiol. Biochim., 1980, 88, 99-105.

Healing, G.; Smith, D. Handbook of pre-clinical continuous intra- venous infusion Taylor & Francis: New York, 2000.

Reilly, R. Variables in animal base research: Part 2. Variability associated with experimental conditions and techniques. ANZCCART News, 1998, Vol.11.

Balcombe, J. P.; Barnard, N. D.; Sandusky, C. Laboratory routines cause animal stress. Contemporary Topics, 2004, 43, 42-51.

Hau, J. S. Handbook of Laboratory Animal Science, Third ed.; CRC Press: Boca Raton, FL, 2010.

Experimental design

No comments

Validity of the findings

no comment

Additional comments

I think a more thorough review of the effect of anesthesia and animal handling is important. In doing so, they may discover their gender are in fact in an earlier report on sex difference in mouse handling.

·

Basic reporting

This paper presents an interesting and useful evaluation of the differential effects of lactate injection in mice. The therapeutic utility of lactate treatment may not be familiar to the general readership of PeerJ and the introduction would benefit from an expanded discussion of the background of this application. In particular, on line 40, the authors mention that lactate administration has therapeutic potential for several neurological diseases. This could lead to an expanded discussion with appropriate references.
A brief discussion of some of the potential side-effects & toxicity potential of lactate therapy would be interesting as well.
One significant issue with the paper is in the presentation of the data in figures 2 and 3. The histogram format is rather cumbersome to review. To present the time course data, it might be more readily interpreted to show a trajectory plot for each treatment. Furthermore, the data could be split into separate figures presented as a series of tiles. For example, Figure 2 could be presented as shown in the attached file.

A simple line diagram or better yet, boxplots of the data could be presented with connecting lines to highlight the trajectory. Separating the PBS and lactate injections, while maintaining a consistent axis range will make the data interpretation much easier. Statistical significance for each plot could be readily represented and differences between plots could be presented in a table format.
The authors could consider the potential effects of differences in total lean muscle mass between the sexes on lactate metabolism.

Experimental design

The experimental design is well constructed and executed. The use of linear mixed effect regression models is appropriate for the studies.
Should be clarified how the 2g/kg body weight dose was determined. It appears to be designed to mimic the effects of high intensity exercise, but this should be stated.

Validity of the findings

Findings are of significant interest and utility to the scientific community involved in research involving lactate metabolism.

Additional comments

Overall a very valuable investigation of lactate metabolism and kinetics using mouse models.

---

## Round 0.2 · accepted · Accept

Thank you for addressing the reviewer concerns and congratulations again.